# Resilience during the COVID-19 pandemic: Associations with changes in burnout and mental well-being among NHS mental health staff in England

Natalia Kika[1]*, Nora Trompeter[1], Danielle Lamb[2], Rupa Bhundia[1], Ewan Carr[3], Brendan Dempsey[2], Neil Greenberg[1], Ira Madan[4], Christopher Penfold[5], Rosalind Raine[2], Simon Wessely[1], Sharon A. M. Stevelink[1]

1 King's Centre for Military Health Research, Institute of Psychiatry, Psychology and Neuroscience, King's College London, London, United Kingdom, 2 Department of Primary Care and Population Health, University College London, London, United Kingdom, 3 Department of Biostatistics and Health Informatics, Institute of Psychiatry, Psychology and Neuroscience, King's College London, London, United Kingdom, 4 Guy's and St Thomas' NHS Trust, London, United Kingdom, 5 National Institute for Health Research Applied Research Collaboration West (NIHR ARC West), University Hospitals Bristol and Weston NHS Foundation Trust, Bristol, United Kingdom

£ These authors contributed equally to this work.
* natalia.kika@kcl.ac.uk

## Abstract

### Background

The pressures of the COVID-19 pandemic placed mental healthcare professionals at an increased risk of burnout and decreased mental wellbeing. Resilience strategies have been put in place as a protective measure, but little is known about how mental wellbeing evolved over time in this group and how resilience affects it. This study aimed to: 1) investigate long-term changes in burnout and mental well-being among mental healthcare professionals working in the National Health Service (NHS) during the COVID-19 pandemic in the United Kingdom (UK); and 2) examine whether baseline resilience levels predicted decreased burnout and increased mental well-being over the course of the pandemic.

### Methods

The study used data from NHS CHECK, a longitudinal cohort study investigating NHS staff mental health and well-being since the COVID-19 pandemic. Clinical mental health staff ($n = 3,289$) who completed self-report measures at three time points (baseline, 6 and 12 months later). Baseline surveys were conducted during the initial pandemic peak (April 2020 – June 2020; $n = 543$), the initial easing of restrictions (July 2020 – September 2020; $n = 1,098$), and the second peak (October 2020 – January 2021, $n = 1,648$).

**Data availability statement:** Due to Research Ethics Committee restrictions, the data set is not publicly available. Requests to access to the de-identified data set can be made to the NHS CHECK team at nhscheck@kcl.ac.uk. Data will be available to researchers who provide a justified hypothesis and structured statistical analysis plan addressing a legitimate research question that is approved by the NHS CHECK Senior Research Team and after the signing of a data sharing agreement. Only deidentified participant data will be provided.

**Funding:** Funding for NHS CHECK was received from the following sources: Medical Research Council (MR/V034405/1); UCL/Wellcome (ISSF3/ H17RCO/C3); Rosetrees Trust (M952 and PGL22/100103); Economic and Social Research Council (ES/V009931/1); NIHR ARC North Thames; NHS England and NHS Improvement; Manolo Blahnik International; Koa Health; Colt Foundation; as well as seed funding from National Institute for Health Research Maudsley Biomedical Research Centre, King's College London, National Institute for Health and Care Research Health Protection Research Unit in Emergency Preparedness and Response at King's College London.

**Competing interests:** The authors have declared that no competing interests exist.

## Results

Mixed model analyses showed that burnout scores increased over time, with higher resilience at baseline predicting lower burnout 6 and 12 months later. However, rises in burnout were most pronounced in the high resilience group. Well-being remained relatively stable over time, with staff with higher resilience at baseline reporting higher well-being over time.

## Conclusions

Resilience was linked with both lower burnout and higher well-being in NHS mental health staff throughout the COVID-19 pandemic. Despite showing steeper increases in burnout, staff with high initial resilience still maintained lower absolute levels of burnout compared to those with lower resilience. Healthcare organizations should consider providing interventions focusing on organizational factors in addition to individual-level resilience-focused support.

## Introduction

The COVID-19 pandemic has posed severe challenges for mental health staff due to staff shortages, increased demand in services, and challenges of remote working [1]. Implementation of infection-control measures in mental health settings resulted not only in changed working conditions but also in reduced patient care [2]. These included cancelled outpatient appointments, reduced group sizes in group therapy, decreased inpatient services, and some hospitals turning psychiatric wards into COVID-wards [3–6]. Thus, the practical limitations in the delivery of mental health care, while the need for mental health services continued to increase due to the additional burden of the pandemic [7], placed mental health staff under increased pressure and at risk of burnout and decreased mental well-being [1]. Critically, some of the issues remain despite the acute stages of the pandemic easing, with an increasing workload and a backlog of referrals for mental health staff as the pandemic has progressed [8]. It is therefore crucial to consider factors that can both reduce the risk of burnout and increase mental well-being among mental health staff. This study will examine the longer-term changes in burnout and mental well-being of mental health staff during the pandemic and consider whether resilience, defined as the ability to bounce back from setbacks during challenging times [9], decreased burnout and increased mental well-being.

Burnout has long been regarded as a critical issue among mental health staff [10] and while it is not classified as a medical condition, it is defined as an occupational syndrome in the International Classification of Diseases (ICD-11) [11]. With the onset of the COVID-19 pandemic there has been an increased focus on burnout and mental well-being among caring professionals due to the pandemic pressures adding to existing high workloads [12]. Further, there are concerns that changes experienced during the pandemic will have long-lasting impacts. A qualitative study of mental health nurses revealed that staff expressed concerns about both the immediate risk

of COVID-19 infection and the pandemic's long-term impact on patients' increasing mental health needs [13]. Similarly, another study found that a fear of infection and a lack of organisational support were determinants of burnout among mental health staff during the pandemic [14]. As such, it is important to consider both the prevention of burnout and the promotion of mental well-being among mental health staff. One common factor examined in the literature regarding this aspect has been psychological resilience [15].

Resilience refers to the ability of individuals to positively adapt to adverse events or experiences and is considered an acquired skill as opposed to a stable trait [16], making it a valuable intervention target. Data collected during the early stages of the pandemic suggest that higher resilience levels were linked to lower depression, stress and anxiety in health-care staff [17]. Similarly, resilience has been linked to lower turnover intention among healthcare staff [18], lower anxiety [19], lower rates of anxiety or depression [20], as well as higher quality of life [21]. Therefore, this cross-sectional research suggests that resilience may be related to lower burnout and higher mental well-being among healthcare staff. Since resilience is partly defined by one's ability to adapt and recover from challenges [22], some initial distress in response to stressors is expected. To fully understand this recovery process, longitudinal studies are necessary to examine how resilience affects long-term adaptation during prolonged stressful events, such as the COVID-19 pandemic. Early research supports this adaptation pattern among healthcare workers [23]. For example, in Spain, while healthcare workers initially showed higher levels of anxiety, depression, and stress than the general public in March 2020, by March 2021, they reported lower levels of these symptoms compared to the general population [24]. A UK study found that healthcare workers' rates of depression and anxiety were four times higher during the pandemic compared to pre-pandemic levels, though workers with higher resilience showed fewer symptoms [25]. However, a systematic review of healthcare workers with moderate to high resilience during the first year of the pandemic suggested that resilience levels may have evolved over time [26]. This highlights the need for long-term studies to specifically track mental healthcare staff's resilience patterns throughout the pandemic.

Emerging longitudinal research supports the existence of varied patterns. Thus, some research demonstrates that mental distress increased over the course of the pandemic among individuals with low or normal levels of resilience, but no change was observed among individuals with high levels of resilience [27]. In contrast, research among healthcare workers in the United States found no significant relationship between resilience at the start of the pandemic (March 2020) and changes in psychological distress after three months [23]. Thus, there is currently no clear evidence that resilience is linked with either decreased burnout or increased mental well-being, and no existing research has examined this prospective relationship among mental health staff whose challenges may be specific to their roles.

Despite the lack of evidence, many organisations quickly adopted resilience interventions during the pandemic to increase staff well-being [28,29]. While initial studies suggest positive outcomes from these interventions [30], a systematic review indicated that there is a lack of evidence to definitively determine their effectiveness [31]. Research shows that workplace factors, not individual traits, were the main drivers of mental health staff burnout during the pandemic [14]. Therefore, focusing on personal resilience instead of improving organizational support and work conditions may be counterproductive. This concern is not new; critics have long argued that resilience-focused interventions shift responsibility to individuals rather than addressing broader organizational and social issues [32]. Moreover, implementing interventions targeting unproven factors is premature, especially when evidence does not support they reduce burnout or improve well-being.

Implementing resilience-based interventions may create additional strain on mental health staff who already lack adequate support systems. Research illustrates this challenge from two perspectives. In Canada, a study of a resilience coaching program showed mental health staff played a vital role in supporting their acute care colleagues during the pandemic [33]. Similarly, in the UK, mental health workers who supported frontline staff reported mixed experiences [34]. While they found supporting colleagues rewarding, they lacked proper training and guidance for this role. Furthermore, these mental health staff faced unsustainable workloads and insufficient support for themselves, particularly when exposed to their colleagues' potentially traumatic experiences [35].

The current study sought to address these critical gaps in the literature by examining long-term changes in burnout and mental well-being among mental health staff working within the National Health Service (NHS) during the pandemic. Specifically, we investigated: 1) were long-term changes in burnout and mental well-being present among mental healthcare professionals working in the NHS during the COVID-19 pandemic in the United Kingdom (UK)?; and 2) did baseline resilience levels predict decreased burnout and increased mental well-being over the course of the pandemic?. To address these research questions, we considered whether resilience at the start of the pandemic was linked with subsequent decreases in burnout and increases in mental well-being after 6 and 12 months.

## Methods

### Sample and procedure

To examine temporal associations between resilience, burnout, and mental wellbeing among mental health staff, we used data from NHS CHECK [36], a longitudinal cohort study of the mental health and well-being of health care staff during the COVID-19 pandemic in England. We chose this study since it measured our outcome variables at three timepoints during the COVID-19 pandemic, and it included participants who identified as mental health staff in the NHS. In brief, the study recruited NHS staff from 18 participating Trusts to participate in a longitudinal study on the effects of the pandemic. All staff (clinical and non-clinical) were invited to complete an online survey between 27/04/2020 and 15/01/2021 (initial pandemic peak). Participants were then re-invited to complete surveys 6 (27/10/2020 until 15/08/2021; initial easing of restrictions) and 12 months (27/04/2021 until 21/02/2022; second peak) after their initial response, with a four-week window to complete it. To minimise participant burden, all participants completed a shorter survey, and had the opportunity to complete an additional, longer survey with further psychosocial measures. Details on the methodology are available in the NHS CHECK protocol [36].

To address the evidence gaps outlined above, we included all staff who were identified as clinical mental health staff. That is, psychologists, assistant psychologists, psychiatrists, psychiatric nurses, or other clinical staff working in psychological services, psychiatry, or secure settings ($n = 6,322$). As the current study used measures that were only included in the long version of the survey, we restricted analyses to participants who completed both the short and long survey at baseline ($n = 3,289$). Mental health staff who completed the long survey at baseline were more likely to be female ($p < .001$, $V = .05$), White ($p < .001$, $V = .11$), and older ($p = .01$, $V = .05$); however, effect sizes were weak, indicating few meaningful differences between participants who only completed the short survey, and those who completed the additional longer version.

### Measures

**Resilience.** Resilience levels were assessed with the Brief Resilience Scale (BRS) [37] at baseline. Participants rated their personal resilience across 6 items on a 5-point scale (1 = *Strongly disagree* to 5 = *Strongly agree*). Three items were positively worded (e.g., *"I tend to bounce back quickly after hard times"*) and three items were negatively worded and reverse scored (e.g., *"I have a hard time making it through stressful events"*). A total score was calculated to reflect overall resilience, with higher scores reflecting higher levels of resilience. The scale showed good internal consistency in the current study at baseline (Cronbach's alpha = .89).

**Burnout.** Burnout was examined through the Burnout Assessment Tool short version (BAT-12) [38], a 12-item version of the original 23-item questionnaire, which covers four dimensions of burnout: exhaustion (e.g., *"at work, I feel mentally exhausted"*), mental distance (e.g., *"I struggle to find any enthusiasm for my work"*), cognitive impairment (e.g., "*when I'm working, I have trouble concentrating"*), and emotional impairment (e.g., *"at work, I feel unable to control my emotions"*). Each statement was rated on a 5-point scale (1 = *never* to 5 = *always*). A mean score was calculated to reflect overall burnout. The scale showed good internal consistency in the current study at all three timepoints (Cronbach's alpha = .91; .89; .92).

**Mental well-being.** Mental well-being was examined using the Warwick–Edinburgh Mental Wellbeing Scale (WEMWBS) [39]. The 14-item scale included statements covering emotional and functioning aspects of mental well-being. Participants rated how frequently each statement applied to them in the previous two weeks (e.g., "*I've been feeling useful*") on a 5-point scale (1 = *none of the time* to 5 = *all of the time*). A total score was obtained by summing all items, with greater scores indicating greater well-being. In the UK, the general population has a mean of 51 with a standard deviation of 7. The scale showed good internal consistency in the current study at all three timepoints (Cronbach's alpha = .93;.94;.95).

**Control variables.** To control for potential confounding variables, we examined the following control variables which were chosen based on existing literature and expert opinion: age, sex, ethnicity, relationship status, length of time since professional registration, support from colleagues/supervisors/family & friends, COVID infection (yes/no).

## Statistical analysis

The analysis plan was pre-registered (https://osf.io/audb8) and data were analysed using Stata 17 [40]. All deviations from this plan are outlined below. To examine the associations between resilience and burnout, as well as mental well-being, we used mixed effect modelling for the two outcome variables separately. To account for the different stages of the pandemic in which the baseline survey was completed, we stratified by time of baseline completion to account for different stages of the pandemic: initial peak (April 2020 – June 2020; $n = 543$), initial easing of restrictions (July 2020 – September 2020; $n = 1,098$), second peak (October 2020 – January 2021, $n = 1,648$). However, as results were largely the same across the three groups, we ran all analyses in the whole sample, controlling for time of baseline completion.

A three-level structure was modelled for each with time-invariant NHS trust at the third level, participant information (i.e., between-subject) at the second level and time-variant repeated measures (i.e., within-subject) at the first level. The analysis first considered the effects of time for each outcome variable. That is, growth modelling of 'Time' (i.e., baseline/6-month follow-up/12-month follow-up) was modelled as a linear, or quadratic trend to determine a pattern of change over-time. For most models a linear trend was significant, except for some of the well-being analyses where neither trend was significant. In these instances, we decided to continue with a linear trend as the most appropriate option.

In the second part of the analysis, we added socio-demographic control variables (e.g., age, sex, education) to the model, one at a time, both as a main effect and interaction with time. The model indices, including AIC, BIC, and χ2 log-likelihood comparisons, were used to determine whether a particular predictor substantially improves the model and should be retained as a control variable. However, we deviated from our pre-registered analyses and retained some control variables even if they did not improve model fit, for consistency between analyses.

Lastly, the final model was fitted by including all relevant control variables and resilience at baseline to the model to determine whether resilience at baseline predicted change in burnout and mental well-being over time. Resilience was included both as a main effect and interaction with time. Missing data was dealt with using maximum likelihood imputation.

Based on previous analysis from the NHS CHECK study, we used sampling weights to account for sampling bias. The sampling procedure used population-level demographic data provided by each participating Trust's HR department as of April 2020, such as the number of employees and the age, sex, and ethnicity composition of the workforce, and a breakdown by job role. This information was used to calculate a response rate within each Trust. Response weights were generated using a raking algorithm based on age, sex, ethnicity, and role, using the full baseline cohort. Missing data in the weighting process were imputed using the 5th nearest neighbour (kNN) algorithm. Finally, individual data was weighted by Trust size, and demographic data (ethnicity, categorical age, role and sex).

## Ethics

Ethical approval for the NHS CHECK study was granted by the Health Research Authority (reference: 20/HRA/2107, IRAS: 282686) and local Trust Research and Development approval. Participants' written consent was obtained at each study wave.

## Results

### Sample characteristics

Table 1 outlines the demographic characteristics of the current sample at baseline. The sample was predominantly female and White. The sample was diverse in terms of job roles, comprised mostly of mental health nurses (unweighted: $n = 1,178$; 35.8%), psychologists (unweighted: $n = 845$; 25.7%), healthcare/nursing assistants (unweighted: $n = 376$; 11.4%), doctors/psychiatrists (unweighted: $n = 295$; 9.0%), and occupational therapists (unweighted: $n = 239$; 7.3%).

### Burnout

A significant linear trend of increasing burnout was observed over time ($p < .001$). Resilience at baseline significantly predicted lower levels of burnout ($B = -0.06$ [$-0.06$; $-0.05$], $p < .001$). Additionally, there was a significant interaction with time ($B = .01$ [.01, .01], $p < .001$), whereby people with higher resilience levels at baseline had the steepest increases in burnout. Simple slope analyses were conducted to probe the interaction further. That is, we tested the significance of three different slopes: low resilience (one or more standard deviations below the sample mean), average resilience (within one standard deviation of the sample mean), and high resilience (one or more standard deviations above the sample mean). Results are outlined in Fig 1. As can be seen, slopes were steepest in the high resilience group indicating that this group reported the greatest increases in burnout over time.

Results stratified by group were similar, resilience at baseline significantly predicted lower levels of burnout in the initial peak group ($B = -0.07$ [$-0.07$; $-0.06$], $p < .001$), the initial easing group ($B = -0.06$ [$-0.08$; $-0.05$], $p < .001$), and the second peak group ($B = -0.05$ [$-0.07$; $-0.04$], $p < .001$). A significant interaction with time was observed in the initial peak group ($B = .01$ [.00, .02], $p = .020$), and second peak group ($B = .01$ [.00, .01], $p = .007$), but not in the initial easing group ($B = .01$ [$-.00$, .02], $p = .154$). Detailed model coefficients are reported in S1 Table.

### Mental well-being

No clear trend regarding mental well-being in relation to resilience was observed over time in the overall sample. Within the initial easing group (July-Sept 2020) there was a significant linear trend for decreasing mental well-being ($p = .002$), however, no significant trend emerged for the initial peak group ($p = .489$) or second peak group ($p = .638$). After inspecting the data, we continued with a linear trend model. As a first step for the analyses, we examined significant covariates. Details regarding the covariate selection process are reported in S2 Appendix. After considering both significant predictors and model fit indices, the final analyses controlled for age, ethnicity, relationship status, time since professional registration, family support, manager support, colleague support, and COVID-19 symptoms.

Resilience at baseline significantly predicted lower levels of mental well-being ($B = 0.78$ [0.66; 0.90], $p < .001$). Additionally, there was a significant interaction with time ($B = -0.08$ [0.66; 0.90], $p < .001$), whereby decreases in well-being were present for those with high resilience only. Simple slope analyses were conducted and are reported in Fig 2.

Results stratified by group were somewhat similar, whereby resilience at baseline significantly predicted lower levels of mental well-being in the initial peak group (April-June 2020; $B = 0.72$ [0.42; 0.24], $p < .001$), the initial easing group (July-Sept 2020; $B = 0.77$ [0.60; 0.94], $p < .001$), and the second peak group (Oct 2020-Jan 2021; $B = 0.79$ [0.64; 0.93], $p < .001$). However, there were no significant interactions with time for any of the three groups and hence no further analyses (i.e., simple slope analyses) were conducted. Detailed model coefficients are reported in S2 Appendix.

## Discussion

This study examined the long-term changes in burnout and mental well-being among NHS mental health staff throughout the COVID-19 pandemic, and their relationship with resilience. Four key findings were observed. First, there was an increase in burnout over time among all clinical mental health staff. Second, higher initial resilience at baseline

**Table 1. Sample characteristics at baseline.**

| | Group | | | |
|---|---|---|---|---|
| | Initial peak Apr-Jun 2020 N (%) | Initial easing Jul-Sep 2020 N (%) | Second peak Oct 2020-Jan 2021 N (%) | Total |
| Sex | | | | |
| Female | 443 (81.58) | 881 (80.24) | 1,321 (80.26) | 2,645 (80.47) |
| Male | 100 (18.42) | 213 (19.40) | 316 (19.20) | 629 (19.14) |
| Ethnicity | | | | |
| White | 464 (85.45) | 1,007 (91.88) | 1,522 (92.80) | 2,993 (91.28) |
| Black | 24 (4.42) | 22 (2.01) | 38 (2.32) | 84 (2.56) |
| Asian | 38 (7.00) | 34 (3.10) | 36 (2.20) | 108 (3.29) |
| Mixed/multiple racial or ethnic groups | 15 (2.76) | 24 (2.19) | 39 (2.38) | 78 (2.38) |
| Age (years) | | | | |
| ≤30 | 116 (21.76) | 216 (20.47) | 309 (19.54) | 641 (20.23) |
| 31–40 | 164 (30.77) | 250 (23.70) | 331 (20.94) | 745 (23.51) |
| 41–50 | 131 (24.58) | 298 (28.25) | 425 (26.88) | 854 (26.95) |
| 51–60 | 94 (17.64) | 238 (22.56) | 420 (26.57) | 752 (23.73) |
| ≥61 | 28 (5.25) | 53 (5.02) | 96 (6.07) | 177 (5.59) |
| Relationship status | | | | |
| Married/Civil partnership | 226 (41.62) | 528 (48.22) | 764 (46.61) | 1,518 (46.32) |
| Co-habiting/In a relationship | 172 (31.68) | 350 (31.96) | 465 (28.37) | 987 (30.12) |
| Divorced/ separated/ widowed | 29 (5.34) | 73 (6.67) | 156 (9.52) | 258 (7.87) |
| Single | 116 (21.36) | 144 (13.15) | 254 (15.50) | 514 (15.69) |
| Time since professional registration | | | | |
| < 1 year | 23 (4.25) | 58 (5.30) | 88 (5.35) | 169 (5.15) |
| 1–2 years | 38 (7.02) | 70 (6.39) | 119 (7.23) | 227 (6.92) |
| 3–5 years | 84 (15.53) | 114 (10.41) | 167 (10.15) | 365 (11.12) |
| 6–10 years | 90 (16.64) | 142 (12.97) | 203 (12.34) | 435 (13.26) |
| 11–20 years | 121 (22.37) | 231 (21.10) | 306 (18.60) | 658 (20.05) |
| 21–29 years | 73 (13.49) | 141 (12.88) | 214 (13.01) | 428 (13.04) |
| ≥30 years | 50 (9.24) | 136 (12.42) | 199 (12.10) | 385 (11.73) |
| Not applicable | 62 (11.46) | 203 (18.54) | 349 (21.22) | 614 (18.71) |
| COVID −19 symptoms | | | | |
| No | 293 (54.77) | 660 (60.94) | 1,073 (66.07) | 2,026 (62.49) |
| Yes | 159 (29.72) | 306 (28.25) | 427 (26.29) | 892 (27.51) |
| Unsure | 83 (15.51) | 117 (10.80) | 124 (7.64) | 324 (9.99) |
| Resilience (mean score, SD) | 21.43 (4.61) | 20.65 (4.64) | 20.83 (4.79) | 20.87 (4.72) |
| Burnout (mean score, SD) | 2.21 (0.63) | 2.26 (0.67) | 2.21 (0.68) | 2.23 (0.67) |
| Mental well-being (mean score, SD) | 45.63 (8.52) | 44.63 (8.75) | 44.40 (9.27) | 44.68 (8.98) |

SD = Standard deviation. The following groups were included under each ethnicity category: White (White English/Welsh/Scottish/Northern Irish/British; White Irish; White Gypsy or Irish Traveller; Any other White background); Black (Black African; Black Caribbean; Any other Black background); Asian (Indian; Pakistani, Bangladeshi; Chinese; Any other Asian background); Mixed/multiple racial or ethnic groups (Mixed White and Black Caribbean; Mixed White and Black African; Mixed White and Asian; Any other Mixed/Multiple ethnic background).

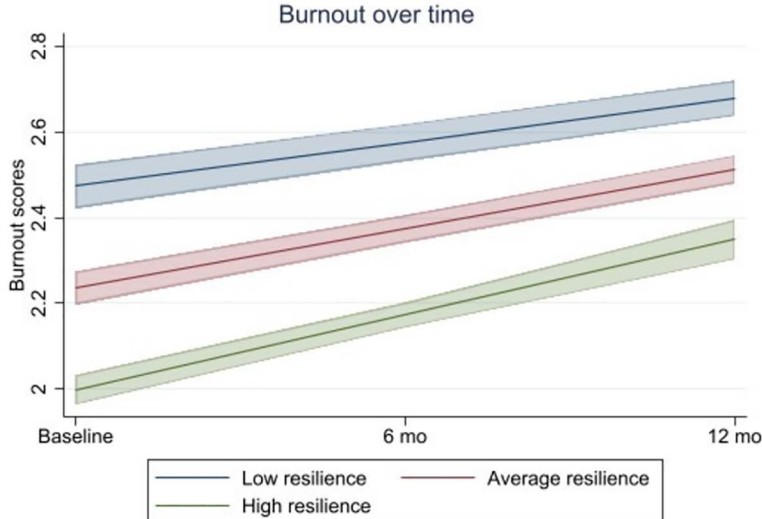

**Fig 1. Burnout scores over a 12-month period with 95% confidence intervals in the overall sample.** Findings are adjusted for time of baseline completion, ethnicity, relationship status, time since professional registration, and Covid-19 symptoms at baseline.

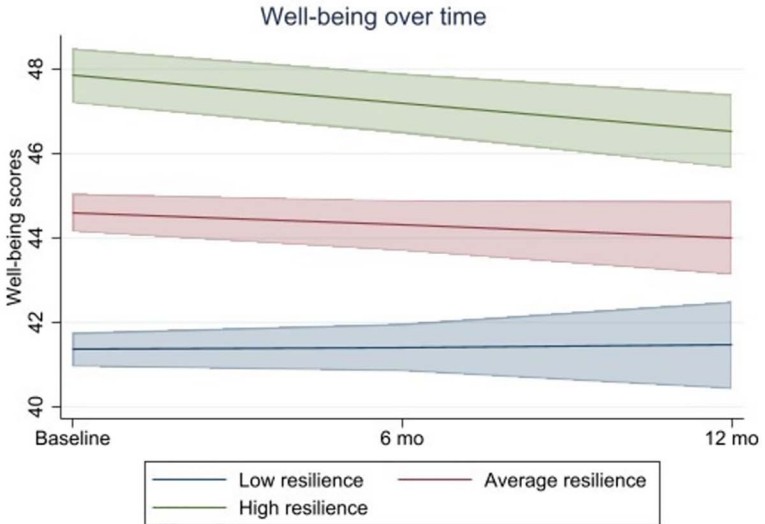

**Fig 2. Well-being scores over a 12-month period with 95% confidence intervals in the overall sample.** Findings are adjusted for time of baseline completion, ethnicity, relationship status, age, time since professional registration, family support, manager support, colleague support, and Covid-19 symptoms at baseline.

predicted lower burnout at 6 and 12 months. Third, staff with higher initial resilience levels showed the steepest increases in burnout over time. Despite this, they remained the least at risk of burnout at 12-month follow-up. Fourth, baseline resilience levels predicted mental well-being trajectories over 12 months. While individuals with lower initial resilience reported consistently lower well-being, those with higher initial resilience showed a decline in well-being over time.

Our longitudinal findings on resilience, burnout, and mental well-being among mental health staff are novel, making direct comparisons with other studies difficult. Although staff with higher resilience showed lower overall burnout, they did not demonstrate the 'bounce back' effect observed in previous cross-sectional studies [24,27]. While cross-sectional studies may have captured temporary decreases in burnout during the pandemic, potentially in between pandemic peaks, our longitudinal findings highlight the prolonged impact of pandemic-related stressors in the workplace, preventing a return to pre-pandemic stress levels. Specifically, these findings suggest that resilience is an important factor in staff burnout and wellbeing, but may not be enough to protect them from increases in burnout and decreased wellbeing during prolonged periods of pressure and stressors such as pandemics. Staff with higher initial resilience showed the largest decline in burnout and well-being scores over time. However, despite this decline, they still maintained better overall scores than those who started with lower resilience. The larger decline may be partially explained by their better initial scores resulting in more room for negative change.

*Because resilience alone did not protect mental health staff from negative changes in burnout and wellbeing throughout the pandemic, our findings suggest that other protective factors remain to be explored. This can be considered through the lens of the job demands-resources (JD-R) framework [41]. The JD-R framework posits that work environments can be explained by a dual process involving two types of job characteristics: job demands, such as workload and work pressure, which require sustained energy and can lead to negative changes in employee wellbeing; and job resources, such as resilience, motivation, and social support, which can buffer any negative effects of job demands and stimulate personal growth. According to the model, resilience would be expected to buffer against increased job demands during the pandemic; however, our findings show this was not the case. The unprecedented job demands experienced during a crisis such as the COVID-19 pandemic may have led to a psychological cost that could not be mitigated by personal job resources such as resilience. Indeed, a recent expansion of the JD-R which accounts for crisis factors in the pandemic found that in emergency contexts, job resources on a social support, leadership and organisational level should be considered in additional to individual factors [42]. During times of increased job demands, such as infection control measures, a sudden shift to telehealth provision, and supporting frontline colleagues with their mental health, resilience alone may not be enough to buffer the negative effects of the increased job demands. In line with the expanded JD-R, our findings corroborate the need for organisational support targeting factors beyond resilience.*

### Implications for practice and future research

Although staff with higher resilience showed better overall outcomes, our findings, in line with the JD-R model suggest that multiple factors affect burnout during and after the pandemic. Strong evidence shows that organizational interventions, such as improved leadership and work-life balance, effectively reduce healthcare worker burnout [43]. Further organizational factors such as adequate staffing, break times, and basic needs—play a more crucial role in staff well-being than individual resilience [44]. Thus, while resilience interventions may be useful for staff with lower resilience levels, there are concerns that focusing on resilience, particularly among those who are already highly resilient, may be counter-productive, and overshadow organisational factors involved in staff well-being. These groups of staff may feel personally responsible for their struggle to cope, having already developed resilience skills yet still feeling increasingly burnt out.

All mental health staff in our sample reported increases in burnout in the first year of the pandemic, suggesting that many continued to struggle beyond just the initial stages. Further longitudinal studies are needed to investigate how these trajectories evolved over time both after the first year and since pandemic eased. These could assess whether the same levels of burnout and decreased mental wellbeing persisted, and whether the pattern found in our study remained the same or evolved among mental health staff beyond the COVID-19 context. The current and future findings can directly inform the support needs of mental health staff, both for any future pandemics, but also in light of the continued pressures and stressors that these staff groups experience beyond the COVID-19 pandemic.

Mental health staff continue to face unique challenges in their profession due to the increased burden on mental health services since COVID-19. Some of the changes implemented in mental healthcare provision during the acute stages of the pandemic, such as the switch to telehealth instead face-to-face settings [45], remain in place to date, despite healthcare professionals in other fields reverting to pre-pandemic ways of working.. The global burden of mental health disorders had already risen more than expected before the pandemic [46], and since COVID-19 this increase has only been accelerated [47]. Therefore, there is a pressing need to continue supporting the wellbeing of mental health staff who may not have had enough time and resources to recover from the acute pressures and burnout during the pandemic, and who are working on meeting the increased demands on mental health services.

Our findings also highlight that despite the focus on frontline healthcare workers' wellbeing during the pandemic, allocation of resources to organisational support of mental health staff is needed to bolster resilience and coping with the consequences of the pandemic and any similar future situations. Support should target not only staff believed to be less resilient, but also those with high levels of resilience, since our findings showed that highly resilient staff can also struggle without adequate resources and support. Future studies could also investigate any differences in burnout and wellbeing trajectories, or the factors affecting them, among sub-groups of mental health staff.

## Strengths and limitations

Our study's strengths include the use of a weighted sample of mental health staff, making them representative of the populations in the participating NHS trusts in England and Wales. The study used longitudinal data, allowing for a unique investigation of changes in burnout and wellbeing beyond a single timepoint. Some limitations should also be noted. As mentioned, data in the current study covered the period up until the second peak of the COVID-19 pandemic. Furthermore, exploring whether a change in resilience subsequently affects a change in burnout and well-being would be a helpful step towards understanding the importance of resilience interventions, considering the evolving workplace factors affecting mental health staff following the COVID-19 pandemic. Although the use of a quantitative cohort study design allowed for conclusions to be drawn across a large sample size of mental health staff, we are not able to gain any insight into the individual experiences of burnout and mental wellbeing among mental health staff, or which other factors could explain and add more context to the longitudinal associations found in our study. Further qualitative studies with this staff group could bolster our findings with accounts of lived experiences of mental health staff.

## Conclusion

In conclusion, this longitudinal study demonstrated a steady increase in burnout and slight decline in well-being among NHS mental health staff throughout the COVID-19 pandemic. While staff with high levels of resilience reported the lowest overall levels of burnout and highest levels of mental well-being, they paradoxically showed the steepest increases in burnout over time. Notably, despite experiencing the greatest increases in burnout, staff with high baseline resilience still maintained lower overall burnout levels than their less resilient colleagues. These findings suggest that while resilience may serve as a protective factor, it alone is insufficient to prevent burnout in prolonged challenging circumstances. Healthcare organizations should consider implementing comprehensive support systems that address both individual and structural factors, rather than focusing solely on resilience-building interventions.

## Supporting information

**S1 Table. Mixed model coefficients.**
(DOCX)

**S2 Appendix. Details regarding data-driven process to select covariates.**
(DOCX)

## Acknowledgments

We wish to acknowledge the National Institute of Health and Care Research (NIHR) Applied Research Collaboration (ARC) National NHS and Social Care Workforce Group, with the following ARCs: North Thames, East Midlands, East of England, South West Peninsula, South London, West, North West Coast, Yorkshire and Humber, and North East and North Cumbria. They enabled the set-up of the national network of participating hospital sites and aided the research team to recruit effectively during the COVID-19 pandemic. This work is independent research supported by the National Institute for Health Research (NIHR) Applied Research Collaboration (ARC) North Thames and the National Institute for Health and Care Research Health Protection Research Unit (NIHR HPRU) in Emergency Preparedness and Response, a partnership between the UK Health Security Agency, King's College London and the University of East Anglia. The views expressed are those of the author(s) and not necessarily those of the NIHR, UKHSA or the Department of Health and Social Care. For the purpose of open access, the author has applied [a Creative Commons Attribution (CC BY) license] [an 'Open Government License'] to any Author Accepted Manuscript version arising.

The NHS CHECK consortium includes the following site leads: Siobhan Coleman, Sean Cross, Amy Dewar, Chris Dickens, Frances Farnworth, Adam Gordon, Charles Goss, Jessica Harvey, Nusrat Husain, Peter Jones, Damien Longson, Paul Moran, Jesus Perez, Mark Pietroni, Ian Smith, Tayyeb Tahir, Peter Trigwell, Jeremy Turner, Julian Walker, Scott Weich, Ashley Wilkie. The NHS CHECK consortium includes the following co-investigators and collaborators: Peter Aitken, Ewan Carr, Anthony David, Mary Jane Doherty, Sarah Dorrington, Rosie Duncan, Sam Gnanapragasam, Cerisse Gunasinghe, Stephani Hatch, Danielle Lamb, Daniel Leightley, Ira Madan, Richard Morriss, Isabel McMullen, Dominic Murphy, Martin Parsons, Catherine Polling, Alexandra Pollitt, Anne-Marie Rafferty, Rebecca Rhead, Danai Serfioti, Chloe Simela, Charlotte Wilson Jones.

## Author contributions

**Conceptualization:** Natalia Kika, Nora Trompeter, Danielle Lamb, Sharon A.M. Stevelink.

**Data curation:** Ewan Carr, Christopher Penfold.

**Formal analysis:** Nora Trompeter, Christopher Penfold.

**Funding acquisition:** Simon Wessely.

**Methodology:** Danielle Lamb, Rupa Bhundia, Neil Greenberg, Ira Madan, Rosalind Raine, Simon Wessely, Sharon A.M. Stevelink.

**Project administration:** Rupa Bhundia.

**Visualization:** Nora Trompeter.

**Writing – original draft:** Natalia Kika, Nora Trompeter.

**Writing – review & editing:** Danielle Lamb, Ewan Carr, Brendan Dempsey, Neil Greenberg, Ira Madan, Christopher Penfold, Rosalind Raine, Simon Wessely, Sharon A.M. Stevelink.

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
