## [Decision Letter · Decision Letter 0]

PONE-D-24-44533Resilience during the COVID-19 pandemic: Associations with changes in burnout and mental well-being among NHS mental health staff in EnglandPLOS ONE

Dear Dr. Kika,

Thank you for submitting your manuscript to PLOS ONE. After careful consideration, we feel that it has merit but does not fully meet PLOS ONE’s publication criteria as it currently stands. Therefore, we invite you to submit a revised version of the manuscript that addresses the points raised during the review process.

We look forward to receiving your revised manuscript.

Kind regards,

Patricia Pariona-Cabrera

Academic Editor

PLOS ONE

Journal Requirements:

Funding for NHS CHECK was received from the following sources: Medical Research Council (MR/V034405/1); UCL/Wellcome (ISSF3/ H17RCO/C3); Rosetrees Trust (M952 and PGL22/100103); Economic and Social Research Council (ES/V009931/1); NIHR ARC North Thames; NHS England and NHS Improvement; Manolo Blahnik International; Koa Health; Colt Foundation; as well as seed funding from National Institute for Health Research Maudsley Biomedical Research Centre, King's College London, National Institute for Health and Care Research Health Protection Research Unit in Emergency Preparedness and Response at King's College London. 

4. In the online submission form, you indicated that due to Research Ethics Committee restrictions, the data set is not publicly available. Requests to access to the de-identified data set can be made to the NHS CHECK team at nhscheck@kcl.ac.uk.Data will be available to researchers who provide a justified hypothesis and structured statistical analysis plan addressing a legitimate research question that is approved by the NHS CHECK Senior Research Team and after the signing of a data sharing agreement. Only deidentified participant data will be provided. 

5. Please amend either the title on the online submission form (via Edit Submission) or the title in the manuscript so that they are identical.

6. Please remove all personal information, ensure that the data shared are in accordance with participant consent, and re-upload a fully anonymized data set. 

Additional Editor Comments :

Dear Authors ,

Please address reviewers's comments to improve the manuscript.

Kind regards,

Dr Patricia Dina Pariona Cabrera

Reviewers' comments:

Reviewer's Responses to Questions

**Comments to the Author**

1. Is the manuscript technically sound, and do the data support the conclusions?

Reviewer #1: Yes

Reviewer #2: Yes

2. Has the statistical analysis been performed appropriately and rigorously? 

Reviewer #1: I Don't Know

Reviewer #2: Yes

3. Have the authors made all data underlying the findings in their manuscript fully available?

Reviewer #1: Yes

Reviewer #2: Yes

4. Is the manuscript presented in an intelligible fashion and written in standard English?

Reviewer #1: Yes

Reviewer #2: Yes

5. Review Comments to the Author

Reviewer #1: Dear Authors,

Well done on a manuscript that addresses an interesting topic on Resilience during the COVID-19 pandemic:

I have made some comments and suggestions which you may wish to consider in revising and refining your paper. I hope these are helpful.

Abstract: In your abstract it would be helpful to conclude with the key implications of your results and significance of your findings to practice and theory. Outline your main contributions.

Introduction: You have made some good points in your introduction regarding the importance of the concepts you are studying. I also suggest that it would be helpful to the reader if you could clearly articulate your research question at the end of your introduction section.

Your final paragraph before you launch into your methods seems to me to belong in the methods section – perhaps under a subheading ‘Sample’.

Just a minor point – I noticed you are missing a punctuation mark after reference [8].

Methodology: In your methodology section, you launch straight into the methodological approach you undertook to collect the data. I suggest you start this section with a justification of your chosen methodological approach – including why this was an appropriate methodological choice to answer your research question. In this section, readers may expect to see citations to researchers who have adopted this approach previously or researchers who have written about the methodological benefits/drawbacks of your selected approach. In particular, I would like to see some more justification of the use of the NHS CHECK study. Justification of methodological approach can help strengthen your manuscript by ensuring you provide adequate reasoning and explanation of your selected methodological approaches. This includes information on methodological approach at a high level (i.e. epistemological approach, choice of qualitative/quantitative, research paradigm i.e. constructivist/positivist etc) as well as the specific method adopted for collecting and analysing data – in your case the NHS Check study. Remember readers may not be familiar with the NHS Check study so it is important to provide some more details and discussion of why it is appropriate in your study context.

Discussion: Good job in the first paragraph summarising your key findings at a high level, but I believe there is room to elaborate on these further. For example, in the section “Despite staff with higher resilience levels showing lower burnout overall, these findings do not support some previous research suggesting that healthcare workers may have ‘bounced back’ from the initial peak of the pandemic [24, 27], although this may be because of the ongoing challenging work environment effectively meaning that staff cannot bounce back to a more usual, less stressful, workplace” - I suggest you further elaborate and explain what this may mean. This will link nicely into a section on theoretical and practical implications of your paper.

My further suggestions:

I believe a section on implications of your study and future research would add value to the paper: You allude to the opportunity for future research stemming from this study, but I encourage you to tease this out a little more beyond what you have provided. Be a little more specific – what studies would you like to stem from this research to add to the contributions of your study? I also suggest you have a section where you discuss the implications of your study and another to discuss your future research recommendations - have separate headings for these. I also suggest you separate the implication section into implications for practice and implications for theory. You may need to add to these sections. This will help highlight your research contribution and value of the study.

Also, I suggest you think resilience post-COVID-19 and justifying why it is still relevant now. I believe it is relevant but stronger justification would help your case and contribution.

Conclusion: It would be great to see a conclusion that summarises the key points of your manuscript.

Theory section: The main drawback of your paper is that you do not appear to use a theoretical lens or theoretical framework through which you discuss your findings. A theoretical lens is helpful when it comes to making sense of how the variables in your study fit together and in this case, impact the wellbeing and mental health of staff in England. For example – you may want to consider some resources such as the Conservation of Resources Theory (as an example) -t here are many other theories you can draw upon which may suit your study objectives better. We would then expect your discussion to interpret the results through your chosen theoretical lens. This will also help you with the theoretical implications section that I am suggesting you add.

Thanks for the opportunity to review your paper!

Reviewer #2: My review did not focus on the methodology and statistical analysis. All my feedback are aimed at simplification to improve clarity because several sentences are long and complex, usually combining multiple ideas. Suggested text were provided to illustrate an example, rather than for adoption. However, feel free to use if you find the suggested edits conveys the message better. All the best on your publication. Cheers!

6. PLOS authors have the option to publish the peer review history of their article (what does this mean? ). If published, this will include your full peer review and any attached files.

**Do you want your identity to be public for this peer review?** For information about this choice, including consent withdrawal, please see our Privacy Policy .

Reviewer #1: No

Reviewer #2: No

---

## [Author Response · Author response to Decision Letter 1]

11 Apr 2025

Dear Dr Pariona-Cabrera,

Thank you for the opportunity to review our manuscript “Resilience during the COVID-19 pandemic: Associations with changes in burnout and mental well-being among NHS mental health staff in England”. We would like to thank the reviewers for their helpful comments and feedback. Please see below an overview of the comments and our responses.

Thank you in advance for your consideration of this resubmission to PLOS ONE, and we look forward to hearing from you.

Yours sincerely,

Natalia Kika, on behalf of the authors

Comments from the Editor:

1. Please ensure that your manuscript meets PLOS ONE's style requirements, including those for file naming

Response: Thank you, we have now updated our manuscript according to the journal’s style requirements.

Response: Thank you for pointing this out. Please see below our updated financial disclosure statement:

Funding for NHS CHECK was received from the following sources: Medical Research Council (MR/V034405/1); UCL/Wellcome (ISSF3/ H17RCO/C3); Rosetrees Trust (M952 and PGL22/100103); Economic and Social Research Council (ES/V009931/1); NIHR ARC North Thames; NHS England and NHS Improvement; Manolo Blahnik International; Koa Health; Colt Foundation; as well as seed funding from National Institute for Health Research Maudsley Biomedical Research Centre, King's College London, National Institute for Health and Care Research Health Protection Research Unit in Emergency Preparedness and Response at King's College London. The funders had no role in study design, data collection and analysis, decision to publish, or preparation of the manuscript.

3. & 4. Data availability

Response: There are ethical restrictions to sharing a de-identified data set for this study since we do not have ethical approval to fully share anonymised data sets containing individual data. Our ethics approval currently only allows study team members to have access to de-identified data sets, unless a data request form has been approved to any individuals outside of the study team. Data requests may be sent to the study team via nhscheck@kcl.ac.uk.

5. Please amend either the title on the online submission form (via Edit Submission) or the title in the manuscript so that they are identical.

Response: Noted, thank you.

6. Please remove all personal information, ensure that the data shared are in accordance with participant consent, and re-upload a fully anonymized data set.

Response: Please see our response to points 3 and 4 as to why we are not able to share a fully anonymised data set.

7. Please include captions for your Supporting Information files at the end of your manuscript, and update any in-text citations to match accordingly.

Response: Thank you, we have no added captions to the end of our manuscript.

Comments from Reviewer 1:

Overall comment: Well done on a manuscript that addresses an interesting topic on Resilience during the COVID-19 pandemic:

I have made some comments and suggestions which you may wish to consider in revising and refining your paper. I hope these are helpful.

Response: We thank the reviewer for their positive comment and helpful suggestions below which we believe have added further value and added clarity to our manuscript.

#1 Abstract: In your abstract it would be helpful to conclude with the key implications of your results and significance of your findings to practice and theory. Outline your main contributions.

Response: We have added a sentence on the study implications to the abstract:

“Healthcare organizations should consider providing interventions focusing on organizational factors in addition to individual-level resilience-focused support.” (pg.1)

#2 Introduction: You have made some good points in your introduction regarding the importance of the concepts you are studying. I also suggest that it would be helpful to the reader if you could clearly articulate your research question at the end of your introduction section.

Response: We thank the reviewer for the positive comment and suggestion on clarifying our research question. We have now updated this as follows:

“The current study sought to address these critical gaps in the literature by examining long-term changes in burnout and mental well-being among mental health staff working within the National Health Service (NHS) during the pandemic. Specifically, we investigated: 1) whether there were long-term changes in burnout and mental well-being among mental healthcare professionals working in the NHS during the COVID-19 pandemic in the United Kingdom (UK); and 2) whether baseline resilience levels predicted decreased burnout and increased mental well-being over the course of the pandemic. To address these research questions, we considered whether resilience at the start of the pandemic was linked with subsequent decreases in burnout and increases in mental well-being after 6 and 12 months.” (pg. 7)

#3 Introduction: Your final paragraph before you launch into your methods seems to me to belong in the methods section – perhaps under a subheading ‘Sample’.

Response: We removed the detailed information on data collection dates for the three pandemic timepoints, and clarified this information in the Methods section (pg. 8; please also see the response for comment #2).

#4 Introduction Just a minor point – I noticed you are missing a punctuation mark after reference [8].

Response: We have now amended this.

#5 Methodology: In your methodology section, you launch straight into the methodological approach you undertook to collect the data. I suggest you start this section with a justification of your chosen methodological approach – including why this was an appropriate methodological choice to answer your research question. In this section, readers may expect to see citations to researchers who have adopted this approach previously or researchers who have written about the methodological benefits/drawbacks of your selected approach. In particular, I would like to see some more justification of the use of the NHS CHECK study. Justification of methodological approach can help strengthen your manuscript by ensuring you provide adequate reasoning and explanation of your selected methodological approaches. This includes information on methodological approach at a high level (i.e. epistemological approach, choice of qualitative/quantitative, research paradigm i.e. constructivist/positivist etc) as well as the specific method adopted for collecting and analysing data – in your case the NHS Check study. Remember readers may not be familiar with the NHS Check study so it is important to provide some more details and discussion of why it is appropriate in your study context.

Response: We have added a sentence to justify the choice of a longitudinal cohort study design to the Methods section:

We adopted a longitudinal cohort study design as we were interested to examine temporal associations between resilience, burnout, and mental wellbeing among mental health staff. We used data from the NHS CHECK study [36], a longitudinal cohort study of the mental health and well-being of health care staff during the COVID-19 pandemic in England and Wales. We chose to use data from this study since it measured our outcome variables at three timepoints during the COVID-19 pandemic, and it included participants who identified as mental health staff in the NHS. (pg. 7)

We also added further information to the ‘strengths and limitations’ section in the Discussion to acknowledge the drawback of using a quantitative over a qualitative approach to address our research question:

Although the use of a quantitative cohort study design allowed for conclusions to be drawn across a large sample size of mental health staff, we were not able to gain any insight into the individual experiences of burnout and mental wellbeing among mental health staff. Further qualitative studies with a smaller sample from this staff group could bolster our findings with accounts of lived experiences of mental health staff since the initial stages of the pandemic. (pg. 18)

#6 Discussion: Good job in the first paragraph summarising your key findings at a high level, but I believe there is room to elaborate on these further. For example, in the section “Despite staff with higher resilience levels showing lower burnout overall, these findings do not support some previous research suggesting that healthcare workers may have ‘bounced back’ from the initial peak of the pandemic [24, 27], although this may be because of the ongoing challenging work environment effectively meaning that staff cannot bounce back to a more usual, less stressful, workplace” - I suggest you further elaborate and explain what this may mean. This will link nicely into a section on theoretical and practical implications of your paper.

Response: We expanded on the meaning of these findings and linked them subsequently with a paragraph on theoretical frameworks which we added to the discussion (pg. 15-16).

#7 Discussion: I believe a section on implications of your study and future research would add value to the paper: You allude to the opportunity for future research stemming from this study, but I encourage you to tease this out a little more beyond what you have provided. Be a little more specific – what studies would you like to stem from this research to add to the contributions of your study?

We have added a section for practical implications and future research, including suggestions for specific studies (pg. 16-17).

#8 Discussion: I also suggest you separate the implication section into implications for practice and implications for theory. You may need to add to these sections. This will help highlight your research contribution and value of the study.

Response: We thank the reviewer for this suggestion. We have now included a paragraph where we interpreted our findings through a theoretical framework, and a separate paragraph for implications for practice, as described in points 6 and 7. The theoretical framework paragraph can be seen in point 10 below.

#9 Discussion: Also, I suggest you think resilience post-COVID-19 and justifying why it is still relevant now. I believe it is relevant but stronger justification would help your case and contribution.

Response: We have added information on some post-COVID-19 implications to the Implications section:

Mental health staff continue to face the unique challenges of the increased burdens on mental health services since the pandemic. Some of the changes implemented in mental healthcare provision during the acute stages of the pandemic, such as the switch to telehealth instead face-to-face settings (ref), remain in place to this date, whereas many other healthcare staff have since reverted to pre-pandemic ways of working. Our findings can therefore inform the baseline wellbeing of mental health staff as they entered the post-pandemic phase. These shed light on the importance of considering the specific challenges they faced, and the consequences of these which may persist in the future. The global burden of mental health issues are expected to rise significantly by 2030 (ref), and the gap between the need and provision of mental health care has only increased since COVID-19 (ref), emphasising the need to continue supporting the wellbeing of mental health staff. (pg. 17)

#10 Discussion: Theory section: The main drawback of your paper is that you do not appear to use a theoretical lens or theoretical framework through which you discuss your findings. A theoretical lens is helpful when it comes to making sense of how the variables in your study fit together and in this case, impact the wellbeing and mental health of staff in England. For example – you may want to consider some resources such as the Conservation of Resources Theory (as an example) - there are many other theories you can draw upon which may suit your study objectives better. We would then expect your discussion to interpret the results through your chosen theoretical lens. This will also help you with the theoretical implications section that I am suggesting you add.

Response: We thank the reviewers for the valuable suggestion to add a theoretical framework to our interpretation of findings. We have now added this to the Discussion (pg. 15-16):

Because resilience alone did not protect mental health staff from negative changes in burnout and wellbeing throughout the pandemic, our findings suggest that other protective factors remain to be explored. This can be considered through the lens of the job demands-resources (JD-R) framework [42]. The JD-R framework posits that work environments can be explained by a dual process involving two types of job characteristics: job demands, such as workload and work pressure, which require sustained energy and can lead to negative changes in employee wellbeing; and job resources, such as resilience, motivation, and social support, which can buffer any negative effects of job demands and stimulate personal growth. According to the model, resilience would be expected to buffer against increased job demands during the pandemic; however, our findings show this was not the case. The unprecedented job demands experienced during a crisis such as the COVID-19 pandemic may have led to a psychological cost that could not be mitigated by personal job resources such as resilience. Indeed, a recent expansion of the JD-R which accounts for crisis factors in the pandemic found that in emergency contexts, job resources on a social support, leadership and organisational level should be considered in additional to individual factors [43]. During times of increased job demands, such as infection control measures, a sudden shift to telehealth provision, and supporting frontline colleagues with their mental health, resilience alone may not be enough to buffer the negative effects of the increased job demands. In line with the expanded JD-R, our findings corroborate the need for organisational support targeting factors beyond resilience.

#11 Conclusion: It would be great to see a conclusion that summarises the key points of your manuscript.

Response: We have now added a conclusion as a separate section and improved the clarity of our summary of findings (pg. 18-19):

In conclusion, this longitudinal study demonstrated a steady increase in burnout and slight decline in well-being among NHS mental health staff throughout the COVID-19 pandemic. While staff with high levels of resilience reported the lowest overall levels of burnout and highest levels of mental well-being, they paradoxically showed the steepest increases in burnout over time. Notably, despite experiencing the greatest increases in burnout, staff with high baseline resilience still maintained lower overall burnout levels than their less resilient colleagues. These findings suggest that while resilience may serve as a protective factor, it alone is insufficient to prevent burnout in prolonged challenging circumstances. Healthcare organizations should consider implementing comprehensive support systems that address both individual and structural factors, rather than focusing solely on resilience-building interventions.

Comments from Reviewer 2:

My review did not focus on the methodology and statistical analysis. All my feedback are aimed at simplification to improve clarity because several sentences are long and complex, usually combining multiple ideas. Suggested text were provided to illustrate an example, rather than for adoption. However, feel free to use if you find the suggested edits conveys the message better. All the best on your publication. Cheers!

Response: We thank the reviewer for their helpful suggestions on improving clarity and simplicity of our sentences. These can be found in the revised manuscript as tracked changes on the following pages:

Abstract – pg.3

Introduction – pg. 4, 5, 6, 7

Methods – pg. 7, 8

Discussion – pg. 15, 16, 17

---

## [Decision Letter · Decision Letter 1]

Resilience during the COVID-19 pandemic: Associations with changes in burnout and mental well-being among NHS mental health staff in England

PONE-D-24-44533R1

Dear Dr. Natalia Kika,

We’re pleased to inform you that your manuscript has been judged scientifically suitable for publication and will be formally accepted for publication once it meets all outstanding technical requirements.

Kind regards,

Patricia Pariona-Cabrera

Academic Editor

PLOS ONE

Additional Editor Comments (optional):

Dear Authors,

Please addresses some minor points:- Please proof read for grammar and ensure you do not start sentences with because where possible.

- I suggest actually wording your research questions as questions, complete with question marks.

---

## [Editor Report · Acceptance letter]

PONE-D-24-44533R1

PLOS ONE

Dear Dr. Kika,

I'm pleased to inform you that your manuscript has been deemed suitable for publication in PLOS ONE. Congratulations! Your manuscript is now being handed over to our production team.

Kind regards,

on behalf of

Dr. Patricia Pariona-Cabrera

Academic Editor

PLOS ONE
